# Acute Functional Adaptations in Isolated Presynaptic Terminals Unveil Synaptosomal Learning and Memory

**DOI:** 10.3390/ijms20153641

**Published:** 2019-07-25

**Authors:** Anna Pittaluga

**Affiliations:** 1Department of Pharmacy, DiFAR, Pharmacology and Toxicology Section, Viale Cembrano 4, 16148 and Center of Excellence for Biomedical Research, University of Genoa, Viale Benedetto XV, 16132 University of Genoa, 16145 Genoa, Italy; pittalug@difar.unige.it; 2IRCCS Ospedale Policlinico San Martino, 16145, Genova, Italy

**Keywords:** synaptosomes, in vitro treatment, in vivo treatment, presynaptic receptors, functional adaptation, transmitter release

## Abstract

Synaptosomes are used to decipher the mechanisms involved in chemical transmission, since they permit highlighting the mechanisms of transmitter release and confirming whether the activation of presynaptic receptors/enzymes can modulate this event. In the last two decades, important progress in the field came from the observations that synaptosomes retain changes elicited by both “in vivo” and “in vitro” *acute* chemical stimulation. The novelty of these studies is the finding that these adaptations persist beyond the washout of the triggering drug, emerging subsequently as functional modifications of synaptosomal performances, including release efficiency. These findings support the conclusion that synaptosomes are plastic entities that respond dynamically to ambient stimulation, but also that they “learn and memorize” the functional adaptation triggered by *acute* exposure to chemical agents. This work aims at reviewing the results so far available concerning this form of synaptosomal learning, also highlighting the role of these *acute* chemical adaptations in pathological conditions.

## 1. Introduction

Along brain slices and cultured neurons, isolated nerve endings (we refer to as synaptosomes) can help to elucidate the cascade of events involved in chemical transmission. Synaptosomes permit studying the mechanism of transmitter exocytosis and dissecting the mechanism of control of this functional parameter, allowing to prove the existence of presynaptic receptors, the activation of which modulates the efficiency of transmitters overflow in the synaptic cleft.

Historically, two main approaches have been used to demonstrate the existence and the role of presynaptic release-regulating receptors in isolated nerve terminals. 

The first one relied on “in vitro” studies carried out to quantify the potency and the efficacy of selected ligands at presynaptic release-regulating receptors controlling the transmitter overflows from synaptosomes isolated from distinct regions of the central nervous system (CNS). The results from these “in vitro” experiments also allowed predicting the composition in subunits of the receptor(s) under study as well as identifying the enzymatic repertoire transducing the receptor-mediated signaling from the outer to the inner side of the synaptosomal membranes [1,2,3,4,5].

The second one consisted of “ex vivo, in vitro” studies from synaptosomes that were isolated from central regions of animals *chronically* administered with selective agonists/antagonists. In this case, the working hypothesis relied on the assumption that, although synaptosomes are largely conserved structures, they develop persistent adaptations when the nerve endings they originate from are *chronically* exposed to chemicals in intact tissue (i.e., as is supposed to occur in “in vivo” studies). These structural/functional adaptations were proposed to rely on the onset of cascade of events (i.e., up- and downregulation of receptor proteins, rearrangement of subunits assembly due to in–out constitutive trafficking [5], phosphorylation of intraterminal proteins, genetic and epigenetic adaptations in protein synthesis) strictly dependent on the nature of the receptors targeted and of the drug administered that became detectable in “ex vivo, in vitro” as changes in the synaptosomal receptor repertoire and/or release efficiency [6,7,8]. Because of the causative correlation linking the *chronic* drug administration and the adaptive modifications, the results from “ex vivo, in vitro” studies were also considered largely confirmatory of the presence and of the functional role of release-regulating presynaptic receptors in selected subpopulations of nerve endings. In the meantime, however, they provided the first functional evidence of the onset of a presynaptic form of “long term memory” in nerve terminals that could be relevant to the mechanisms of synaptic plasticity and that should be taken into consideration when analyzing the impacts of therapeutics on chemical transmission [9,10]. 

Important progress in the use of isolated nerve endings came from the observations that, in addition to the abovementioned “in-vivo, long term memory”, synaptosomes also retain functional changes when *acutely* exposed “in vitro” to receptor ligands/enzyme modulators. Very interestingly, and in a way unexpectedly, it was demonstrated that the *acute* exposure of synaptosomes to chemicals primes these particles, leading to functional modifications that persist beyond the washout of the triggering agent and that emerge subsequently as changes in release efficiency. On the whole, these observations seemed predictive of the onset of a form of *acute* “in vitro” synaptosomal memory [11]. Starting from this first observation, data accumulated during the last decade were well consistent with this hypothesis. These works also unveiled the onset of persistent adaptations in synaptosomes following the *acute* “in vivo” administration of ligands targeting selected receptors/transporters. Again, the synaptic adaptations were retained by the “ex vivo, in vitro” synaptosomes and were detectable in “in vitro” studies.

The present review aims at reviewing these observations, proposing the hypothesis that synaptosomes can “learn and memorize” *acute* stimulations and could be useful to study the *acute* plastic modifications of the presynaptic component of synaptic connection in the CNS. 

## 2. Synaptosomes to Study Transmitter Release

At chemical synapsis, the release of neurotransmitters is fundamental to dictate the strength and efficiency of the synaptic contacts. In general, the release of neurotransmitters is triggered by action potentials that invade nerve terminals, permitting the influx of positive charges through the Na^+^ and Ca^2+^ channels in plasmamembranes and the consequent activation of intraterminal Ca^2+^-dependent pathways leading to exocytosis [12,13,14,15]. In addition to the massive activation of nerve plasmamembranes, also local depolarization caused by the in–out movements of ions elicited by ionotropic receptors (i.e., the AMPA and the NMDA receptors [16,17,18]) as well as by electrogenic membrane transporters [19,20], can drive transmitter exocytosis. Notably, metabotropic receptors also participate in defining the efficiency of transmitter exocytosis, but, differently from the ionotropic receptors, they preferentially modulate intraterminal enzymatic pathways, finely tuning, rather than eliciting, transmitter exocytosis [21,22,23].

In the CNS, the organization of the synaptic structures makes it very difficult to dissect the presynaptic component of the chemical transmission from the postsynaptic one, especially in “in vivo” studies or in “in vitro” studies carried out with tissue preparations with a higher level of complexity, such as slices or cultured cells. This complexity, however, is significantly reduced when using synaptosomes [24], a simplified tissue preparation that permits deciphering the presynaptic events [13]. 

During the last 70 years, studies with synaptosomes allowed describing: (i)the mechanisms of storage and of release of transmitters in selected CNS regions [25,26,27];(ii)the existence and the functional role of presynaptic release-regulating receptors [10,28];(iii)the functional and molecular adaptations of the presynaptic releasing machinery and of the presynaptic receptors elicited by “in vivo” chronic drug administration [8,9,12];(iv)the functional and molecular presynaptic changes elicited by genetic manipulation in animals [29,30].

## 3. Synaptosomes…

Synaptosomes are pinched-of nerve terminals isolated from synaptic boutons and from varicosities of axonal processes. First described by Gray and Wittaker in 1962 [24], these particles represent a simplified model to study the mechanism of uptake, storage, and release of classic transmitters and the presynaptic processes that regulate these events. Synaptosomes retain the functional features of the structures they originate from, permitting to describe their main characteristics. They possess native enzymes, transporters and receptor proteins, maintaining the complexity of the organization of the presynaptic component of chemical synapsis. 

The transporters located in synaptosomal plasmamembranes take up the transmitter from the external biophase and store it in the vesicular pool, from which the transmitter is subsequently released upon the application of a depolarizing stimuli to the isolated nerve endings [31]. The specificity of the expression of the carrier in the synaptosomal plasmamembranes assures the selective labeling of defined subpopulations of nerve endings, respecting the geometry of the neuronal network. This selective labeling makes it possible to distinguish among selected subpopulations of nerve terminals, despite the heterogeneous composition of the synaptosomal suspension [32,33] and, as synaptosomes possess the intraterminal machinery that permits the release of transmitter in an exocytotic manner [34], to associate the stimulus-evoked release of a specific transmitter to the corresponding synaptosomal subpopulations, regardless of the percentage of that synaptosomal subfamily with respect to the entire population. It is worth stressing that the composition of the synaptosomal suspension can vary even if preparation originates from the same CNS regions. Variability is an unavoidable experimental limitation that depends on several parameters (including the animals, the multistep technical approach, and the ability to manage the processes) that affect the efficiency of the isolation and purification of the synaptosomal fractions (i.e., the number of synaptosomal particles isolated in each preparation). Another aspect that might influence the results from synaptosomes is the contamination of the synaptosomal suspension due to the presence of a certain percentage (about 3% to 5%) of particles that originate from astrocytes (we refer to them as gliosomes [23]). The low percentage of this contamination together with the selective labeling of the isolated nerve endings, however, minimizes the signaling due to the glial fractions in both the functional studies and the biochemical analysis of the synaptosomal proteins.

It is worth reminding that in some cases, synaptosomes retain fragments of the postsynaptic membranes. The impact of this component in release studies is, however, irrelevant, since the possibility that in superfusion, endogenous substances originating from the postsynaptic side can retrogradely affect the release efficiency at the presynaptic side is minimized by the continuums flowing up–down of the medium that quickly removes any endogenous compounds, preventing their retrograde diffusion from the postsynaptic to the presynaptic side of the synaptic contact (see below).

Finally, synaptosomes possess presynaptic receptors, whose activation controls most of the synaptosomal activities, including transmitter release. Activation of presynaptic receptors tunes the distribution of the ionic charges at the two sides of the membranes, as well as the functions of intraterminal enzymatic pathways controlling phosphorylative processes and second messenger productions. On the whole, these events promote or impede the formation of the SNARE complex and, as a direct consequence, control the vesicular transmitter exocytosis. 

## 4. …And Their Up–Down Superfusion to Monitor Transmitter Release

As to the quantitation of transmitter release, in 1974, Raiteri and colleagues [35] described an experimental approach called the “the up-down superfusion of a thin layer of synaptosomes” (reviewed by [13]), which has become the method of choice for the identification and the functional characterization of defined subfamilies of nerve endings. 

The main feature of this experimental approach is the continuous up–down superfusion with a physiological solution of a monolayer of synaptosomes stratified at the bottom of superfusion chambers. The up–down flowing of the superfusion medium assures the rapid removal of any substances at the outer side of synaptosomes, impeding their re-uptake in the superfused particles. The endogenous molecules released are rapidly removed and cannot interact with receptors located on bystander particles. Furthermore, as already anticipated, the continuum superfusion also prevents retrograde signaling from postsynaptic membranes to the presynaptic ones. 

This experimental approach allows distinguishing between drugs that modulate directly the release of transmitter (i.e., agonists acting at presynaptic release-regulating receptors) and drugs that indirectly influence it, by interfering with the mechanism of the uptake (i.e., catecholamine uptake inhibitors which indirectly augment the amount of catecholamine and indolamine availability in the synaptic cleft) and that cannot modulate transmitter overflow in superfusion studies. 

Using this approach, it was possible to describe and to pharmacologically and functionally characterize several receptors presynaptically located on nerve endings, improving our knowledge of the molecular events on the basis of the efficacy of a large part of therapeutics [23,34,36,37,38,39].

## 5. Presynaptic Release-Regulating Receptors

The finding that nerve terminals possess presynaptic receptors controlling the transmitter exocytosis introduced a new concept in the synaptic transmission. Once considered exclusively unidirectional (from the presynaptic to the postsynaptic component of the chemical synapses), this notion introduced the concept that the chemical information at chemical synapsis can also proceed retrogradely, from the synaptic cleft to the presynaptic component, modifying the efficiency of the chemical transmission [7]. 

The role of presynaptic release-regulating receptors in controlling the strength of the synaptic connection made it imperative to classify them pharmacologically and functionally. In this context, the superfusion of purified synaptosomes is one of the most appropriate models to correctly address the question, since it permits a direct correlation between the occupation of the receptor and the functional output. The results from these studies unveiled that the presynaptic release-regulating receptors are a heterogeneous ensemble, consisting of entities that, based on their location and their functions, can be further subclassified in “autoreceptors” and “heteroreceptors” [10]. 

Presynaptic autoreceptors are those receptors that are activated by the transmitter that is under the control of the receptors themselves. They mediate a mechanism of feedback regulation of the release of their natural activator, by preferentially inhibiting it. Differently, heteroreceptors are sensitive to transmitter that are mainly released by adjacent nerve terminals and that are other than the neuron’s own transmitter. Both auto- and heteroreceptors exert their feedback mechanisms of control of transmitter release when exocytosis is elicited by a low to intermediate frequency of stimulation, reflecting their physiological relevance in controlling chemical transmission. 

The pharmacological and functional characterization of presynaptic release-regulating auto- and heteroreceptors was preferentially accomplished in “in vitro” studies with synaptosomes in superfusion by quantifying the potency and the efficacy of selective receptor ligands (either orthosteric and allosteric agonists and antagonists [23,34,40,41,42,43]) as well as antibodies recognizing the outer sequences of the receptor subunit proteins [39,44] that act as modulators of the receptor-mediated control of transmitter release [45]. 

The efficiency in either activating or reducing the receptor-mediated events permits comparative analysis of the pharmacological profiles of ligands but also confirms the existence of the presynaptic release-regulating receptor in the synaptosomal preparation. The ability of selected ligands to modulate transmitter exocytosis could also suggest the participation of a selected subunit to the assembly of the receptor under study, allowing to predict the receptor subunit composition [39,44]. In general, however, these conclusions require further studies to be confirmed, using nonfunctional approaches including immunological experiments aimed at demonstrating the presence of the subunit proteins in the synaptosomal subpopulation under study [46,47,48,49]. As to this possibility, it is worth stressing that subfractioning techniques are available that allow separating the presynaptic component of synaptosomes from the postsynaptic ones [5,20], permitting to quantify the amount of receptor proteins at the two sides of the synaptic contacts. 

## 6. In Vivo Chronic Drug Administration and “Ex Vivo, In Vitro” Persistent Synaptosomal Adaptations

Often, the “in vitro” studies aimed at the characterization of presynaptic receptors were paralleled by “ex-vivo, in vitro” analysis carried out with synaptosomes isolated from CNS regions of animals *chronically* administered with orthosteric/allosteric receptor ligands as well as carrier blockers, to tune “in vivo” the activity of the presynaptic receptors. In addition to confirming the existence of the presynaptic receptors, these studies also aimed at unveiling drug-induced functional and structural adaptations at the presynaptic component of the synapsis. The results from these experiments suggested that the *chronic* administration of receptor ligands/carrier modulators elicit structural and functional remodeling of the synaptic proteasome. Interestingly, these adaptations emerged in “ex vivo, in vitro” studies as a functional adaptation of the presynaptic receptor repertoire, as suggested by the changes in the potency and the efficiency of the ligands acting at these proteins in synaptosomes isolated from selected CNS regions of the treated animals. It is therefore not surprising that this approach was also applied to confirm the involvement of a selected subpopulation of presynaptic terminals in the synaptic network [6,7,8,9,34,50,51,52].

As far as the presynaptic receptors are concerned, the results from the “ex vivo, in vitro” studies unveiled that, independently on how the receptor ligand controls transmitter exocytosis, auto- and heteroreceptors respond differently to the *chronic* administration of selective agonists. Autoreceptors preferentially downregulate when activated *chronically* by agonists and upregulate following *chronic* deprivation of the endogenous transmitter. Differently, most of the heteroreceptors are refractory towards the drug-evoked adaptation, and, in most cases, they did not undergo up- or down-sensitization after long-lasting exposure to selective receptor ligands [6,34]. 

The molecular events accounting for the “in vivo” drug-induced functional modifications were so far poorly investigated and deserve further studies. Nonetheless, whatever the mechanisms involved, the drug-induced adaptations elicited by the “in-vivo” *chronic* administration of receptor ligands provide the first demonstration that nerve terminals can undergo a form of presynaptic “long term memory” that emerges subsequently in “ex vivo, in vitro” studies as functional changes in the synaptosomal receptor repertoire and/or release efficiency. 

Interestingly, in most of these studies, the impact of the *acute* drug administration on the same functional parameters was rarely, if ever, analyzed, based on the assumption that only the continuous “in vivo” drug delivery could assure the onset of persistent adaptations in nerve endings. 

## 7. “In Vitro” Persistent “Presynaptic Adaptation” in Synaptosomes

Important progress in the knowledge of the dynamic adaptability of isolated nerve endings emerged in 2003, when Bailey and colleagues provided the first evidence of the induction of a chemical form of long-term depression (cLTD) in isolated presynaptic terminals [11]. The study was an extension of a previous investigation showing that the elevation of the intracellular cyclic guanosine monophosphate (cGMP, due to the blockade of the type V phosphodiesterase, PED5) concomitant to the inhibition of the protein kinase type A (PKA), elicited a form of chemical LTD (cLTD) at Schaffer collateral-CA1 and at mossy fiber-CA3 synapses. The cLTD was unaffected by preventing postsynaptically the synaptic transmission; furthermore, the administration of both the enzymes modulators at the postsynaptic level did not trigger a comparable signaling. Based on these observations, the authors proposed that the cLTD in hippocampal slices was preferentially a presynaptic event [53]. Accordingly, in 2003, the authors demonstrated that the exposure of rat hippocampal synaptosomes in superfusion to the mixture of the enzyme inhibitors caused a prolonged inhibition of glutamate release efficiency from these particles, which appeared well consistent with the LTD detected in slices (Table 1). 

The novelty of this study was the timing of the experiments applied. The drug-induced presynaptic adaptation in superfused synaptosomes was detected despite the triggering conditioning drugs being removed far in advance. In particular, synaptosomes were superfused with a Kreb’s ringer containing the two enzyme inhibitors, then replaced (again for 20 min) with drug-free medium and finally challenged with a mild depolarizing stimulus (25 mM KCl for 30 s) to elicit glutamate exocytosis. Despite the removal of the triggering agents and the washout period of 20 min, the depolarization-evoked release of glutamate from treated synaptosomes was significantly reduced when compared to untreated particles. 

The authors concluded that the “in vitro” exposure of synaptosomes to both the enzyme modulators caused a persistent adaptation in the isolated nerve endings, which persisted over time and emerged as reduced transmitter exocytosis when synaptosomes were subsequently depolarized. The complex, multistep cascade of events accounting for the dynamic long-lasting functional modification in synaptosomes remains so far unexplored. Nonetheless, the results were particularly relevant as they first suggested that:(i)synaptosomes are dynamic structures that can develop functionally relevant adaptations when exposed *acutely* “in vitro” to external chemical stimuli;(ii)the *acute* chemical-induced synaptosomal adaptations persist well beyond the physical occupancy of the selected receptor binding site(s), and possibly rely on a drug-induced cascade of events that primes persistently the functions of the isolated nerve endings.(iii)the timing of the onset of the synaptosomal adaptation (about 20 min) tends to exclude that genetic and/or epigenetic modifications can underlie these adaptations, provided that synaptosomes possess the repertoire permitting such a kind of events.

About a decade later, a comparable form of “in vitro” chemically-induced long-lasting “presynaptic memory” in nerve endings was described. In this case, functional adaptations were elicited by exposing in superfusion synaptosomes from the nucleus accumbens to nicotine and/or nicotinic agonists before the application of a mild depolarization elicited by the transient activation of ionotropic glutamate receptors. Synaptosomes were pretreated with standard Kreb’s solution containing nicotine itself or nicotine agonists (at concentrations unable to modify on their own transmitter release) and thereafter challenged with NMDA in the presence of glycine to activate presynaptic release-regulating NMDA receptors and/or AMPA to activate non-NMDA receptors. Again, drug-induced long-lasting adaptation emerged as changes in the efficacy of transmitter exocytosis, measured in this case as release of dopamine elicited by the presynaptic release-regulating ionotropic glutamate receptors [32,54,55] (Table 1).

The study originated from data in literature showing that the dopaminergic nerve endings from the nucleus accumbens possess nicotinic receptors that coexist and cooperate with presynaptic release-regulating NMDA and AMPA receptors [38,56,57,58]. When concomitantly activated, nicotinic receptors were synergic to both the AMPA and the NMDA-mediated releasing activity, reinforcing it [16,56]. The impact of nicotine on ionotropic glutamate receptors, however, drastically changed when synaptosomes were pre-exposed to the cholinergic agonist. Mario Marchi and his colleagues [32] described that the transient activation of nicotinic receptors primes the subunit composition of both the ionotropic glutamate receptors and, consequently, their efficiency in controlling dopamine release. In both cases, pre-exposure of synaptosomes caused a rapid internalization of selected receptor subunits (the GluN2B subunit for the NMDA receptors and the GluA2 subunit for the AMPA receptors), then modifying the number and/or the Ca^2+^ permeability of the ionic channels associated to both the presynaptic release regulating ionotropic glutamate receptors, hampering dopamine exocytosis. 

It is known that AMPA and NMDA receptors undergo rapid constitutive trafficking (which occurs within few minutes; see [5] and references therein) that would account for the functional changes in nicotine-treated particles. This event, however, would quickly recover because of the activation of constitutive compensatory mechanism(s) of insertion of new receptors to re-equilibrate their physiological expression in synaptosomal plasmamembranes. The data from the release experiments showing that nicotine-induced changes to the insertion of AMPA and NMDA receptor subunits in synaptosomal plasmamembranes persist during time would suggest that the kinetic of endocytosis of the glutamate receptor subunits is accelerated in treated synaptosomes, but, concomitantly, that the constitutive mechanisms of restoration of the glutamatergic receptor subunits insertion is slowed down.

More recently, another form of “in vitro” drug-induced, long-lasting presynaptic adaptation was described to occur in rat hippocampal synaptosomes. In this case, the presynaptic adaptation was induced by exposing synaptosomes to low nanomolar concentration of CXCL12, a chemokine that is a marker of central inflammation [59]. CXCL12 is the natural ligand of the CXCR4 receptors, a chemokine receptor that coexists and functionally cross-talks with presynaptic NMDA receptors in both glutamatergic and noradrenergic rat hippocampal nerve endings [60]. CXCL12 acting at CXCR4 receptors cannot release glutamate and noradrenaline on its own but sinergically potentiates the NMDA-evoked release of both transmitters when the CXCR4 and NMDA receptors are concomitantly exposed to the respective agonists. CXCL12-induced positive modulation, however, turns to inhibition when synaptosomes are exposed transiently (20 min) to the chemokine before the application of the depolarizing stimulus (the NMDA/glycine challenge). In this case, the loss of release efficiency relied on changes in the phosphorylation of the GluN1 subunit in Serine 896 [61] (Table 1).

To briefly resume, the results described in this paragraph provide evidence that synaptosomes are plastic entities that can undergo persistent functional and structural “in vitro” modifications when exposed to chemicals, suggesting that they can memorize ligand-induced signaling, then modifying their functional outcomes. In some cases, the mechanisms at the basis of these plastic adaptations have been in part elucidated and suggest the involvement of altered receptor in–out trafficking in synaptosomal plasmamembranes and/or changes in the level of phosphorylation of the receptor subunits. Further investigations are, however, required to better address the cascade of events accounting for the duration of these adaptations in synaptosomes.

## 8. Acute “In Vivo” Drug Administration Elicits Persistent Presynaptic Adaptation in Synaptosomes

In 2007, a study was carried out to evaluate whether and to what extent “in vivo” administration of classic antidepressants (i.e., reboxetine and fluoxetine) could affect the exocytosis of noradrenaline and serotonin from rat hippocampal synaptosomes [8]. 

An interesting observation of the study was that the ”in vivo” *acute* administration of reboxetine almost halved the “ex-vivo, in vitro” exocytosis of both transmitters in hippocampal synaptosomes, while *acute* ”in vivo” fluoxetine administration significantly reduced the “ex-vivo, in vitro” exocytosis of serotonin, leaving unmodified the “ex-vivo, in vitro” KCl-evoked release of noradrenaline. In an attempt to find a rationale, it was proposed that the “in vivo” antidepressant-induced modification of the “ex vivo, in vitro” amine exocytosis could represent the consequence of the transient “in vivo” blockade of the noradrenergic and the serotonergic transporters that in turn could have primed both the noradrenergic and the serotonergic terminals, affecting their efficiency in releasing the two transmitters. It was speculated that both the noradrenaline and serotonin reuptake inhibitors could have augmented the extracellular concentration of the two amines in the synaptic cleft, indirectly favoring the activation of presynaptic auto- and heteroreceptors (Figure 1). 

As to this possibility, it is worth reminding that noradrenergic nerve terminals are endowed with inhibitory α_2_ autoreceptors [62,63,64,65], while serotonergic terminals possess inhibitory 5-HT_1B_ autoreceptors [66,67]. Interestingly, inhibitory α_2_ adrenergic heteroreceptors also exist in serotonergic terminals [68], while data supporting the presence of 5-HT heteroreceptors regulating the release of noradrenaline in noradrenergic terminals are lacking [69]. Although the possibility exists that “in vivo” fluoxetine and reboxetine could also target other membrane targets [70], the distribution of the auto- and the heteroreceptors in noradrenergic and serotonergic appeared well consistent with the functional adaptations observed in hippocampal synaptosomes from rat administered *acutely* with the two antidepressants. Actually, the “in vivo” activation of both the α_2_ autoreceptors and the α_2_ heteroreceptors that would follow the *acute* administration of reboxetine could account for the reduced exocytosis of both noradrenaline and serotonin, while, based on the data so far available in the literature, serotonin-induced activation of serotonergic receptors only occurs at serotonergic terminals, since noradrenergic terminals do not possess serotonin release-regulating heteroreceptors.

In an attempt to confirm the hypothesis that the *acute* “in vivo” activation of presynaptic α_2_ receptors can cause “ex vivo, in vitro” adaptation in synaptosomes isolated from the hippocampus, animals administered *acutely* with desipramine were concomitantly and *acutely* injected with yohimbine (a selective α_2_ receptors antagonist). In accordance with expectations, the “in vivo” administration of the antidepressant primed noradrenergic nerve endings and adaptation emerged in “ex vivo, in vitro” release experiments as a drastic reduction of the K^+^-evoked release of NA when compared to control [71]. The “ex vivo, in vitro” adaptation of NA exocytosis, however, was no more detectable in hippocampal synaptosomes from animals administered with desipramine plus yohimbine. [71] (Table 2). 

To conclude, the *acute* administration of a noradrenaline uptake blocker causes the rapid onset of persistent adaptive changes in hippocampal noradrenergic synaptic boutons that are retained by ex-vivo nerve endings, after isolation from hippocampus homogenates and washing with drug-free solutions and that emerged in “ex vivo, in vitro” study as reduced exocytosis efficiency. To note, the onset of the functional long-lasting presynaptic adaptation strictly relies on the activation of the orthosteric binding site of the α_2_ receptors, as suggested by the efficacy of the co-administration of yohimbine in preventing the presynaptic event. Interestingly, the “in vivo” *chronic* administration of a noradrenaline transporter blocker did not cause significant changes in the efficiency of noradrenaline exocytosis in “ex vivo, in vitro” experiments, possibly because of the downregulation of the α_2_ receptors due to the prolonged increased availability of the extracellular endogenous amine elicited by the drugs [8]. Nonetheless, the reboxetine-induced “ex vivo, in vitro” induction of serotonin exocytosis efficiency was still detectable after chronic administration of the carrier blocker, consistent with the view that heteroreceptors do not desensitize following prolonged exposure of the natural ligand. 

The mechanisms involved in the downregulation of the presynaptic release-regulating noradrenergic and serotonergic receptors are so far unknown, but the possibility exists that in this case, epigenetic and/or genetic events could participate to the changes in release efficiency. Further studies are required to address these aspects.

## 9. Acute “In Vivo” Drug-Induced Adaptation at Glutamatergic Nerve Endings: Implications in Pathological Conditions 

Glutamatergic nerve endings possess both ionotropic and metabotropic presynaptic release-regulating receptors, whose activation affects differently glutamate exocytosis. In general, ionotropic autoreceptors (both NMDA and non-NMDA) elicit the release of glutamate in the absence of concomitant depolarizing stimuli. Differently, metabotropic glutamate receptors (mGlu) finely tune this event when concomitantly added to a depolarizing stimulus. In particular, mGlu receptors belonging to group I permit and reinforce glutamate overflow, while the mGlu receptors belonging to the II (namely, mGlu2/3 receptors) and III group (in particular, mGlu4 and mGlu7 receptor subtypes) inhibit it [39,43].

The hypothesis that the “in vivo” *acute* activation of presynaptic autoreceptors could prime nerve endings causing persistent modification at the presynaptic component of chemical synapsis was confirmed also when studying the role of mGlu2/3 autoreceptors on glutamate exocytosis in both cortical and spinal cord nerve endings of mice. The mGlu2/3 autoreceptors controlling the glutamate exocytosis from the two synaptosomal subpopulations differ in term of affinity and efficacy of selective mGlu2/3 agonists/antagonists, compatible with the conclusion that they represent different mGu2/3 receptor subtypes [72].

In line with the proposed onset of synaptic adaptation due to the *acute* administration of agonists, the *acute* “in vivo” injection of LY379268, a broad spectrum mGlu2/3 agonist, caused persistent modifications to both the cortical and the spinal cord glutamatergic nerve endings that were reminiscent of the presynaptic inhibitory control exerted by mGlu2/3 autoreceptors in these terminals. Actually, “in vitro” glutamate exocytosis evoked by a mild depolarizing stimulus from the synaptosomes isolated from both the CNS regions (i.e., the cortex and the spinal cord) of the “in vivo” treated animals was significantly reduced when compared to that from control untreated animals [71,72,73,74,75].

Again, the hypothesis that synaptosomes could have “memorized” the consequences due to the “in vivo” *acute* activation of the presynaptic autoreceptor agonist well supported the “ex vivo, in vitro” observations.

Multiple sclerosis (MS) is an autoimmune inflammatory disease with unknown etiology. Overt demyelination and inflammation develop in selected regions of the central nervous system (CNS). Comparable histological hallmarks are also detected in the CNS of mice suffering from experimental autoimmune encephalomyelitis (EAE). In addition to inflammation and demyelination, a diffuse loss of synaptic contact, axonal pruning, and astrocytosis was also reported to occur in the CNS of patients suffering from MS and in EAE mice. These observations led to proposing the term “synaptopathy” to describe the demyelinating pathology [76]. As to the synaptic defects, several observations in the last decade supported the onset of altered glutamate release efficiency from synaptosomes isolated from selected CNS regions of EAE mice at different stages of disease [77]. 

To investigate whether the “in vivo” mGlu2/3-induced chemical presynaptic adaptation also occurs in pathological conditions typified by altered glutamatergic innervation, the same experimental paradigm used above in healthy mice was applied to animals suffering from EAE [72]. The data available in the literature demonstrated that the efficiency of glutamate exocytosis from cortical nerve endings is drastically reduced in EAE mice when compared to controls, possibly because of the significant reduction of the intraterminal cyclic Adenylyl cyclase-mediated pathway. Differently, glutamate exocytosis from the spinal cord synaptosomes from EAE mice significantly increases when compared to control.

The *acute* intraperitoneal injection of LY379268 in EAE mice at the acute stage of disease failed to affect the glutamate exocytosis from cortical nerve endings. Since the mGlu2/3 autoreceptors negatively couple adenylyl cyclase-dependent pathways, it was proposed that the reduced efficiency of this enzyme in the glutamatergic cortex of EAE mice at that stage of disease could mainly account for the loss of function of the “in vivo” mGlu2/3 agonist to prime cortical glutamatergic nerve endings. Differently, “in vivo” *acute* LY379268 efficiently reduced the “in vitro” glutamate exocytosis from superfused spinal cord synaptosomes [72]. This observation is particularly relevant from a therapeutic point of view, since clinical observations suggest that MS patients suffer from a reduction in the production of N-acetyl aspartyl glutamate (NAAG), an endogenous dipeptide that acts as a full agonist at mGlu2/3 autoreceptors in the spinal cord. Accordingly, either mGlu2/3 receptor agonist as well as inhibitors of the glutamate carboxypeptidase II (GCPII), i.e., the enzyme catalyzing the cleavage of endogenous NAAG to N-acetyl aspartate and glutamate, were recently proposed to be beneficial for the course of the disease. The data described above could in part give the rationale to the use of NAAG modulators for the cure of MS.

## 10. Concluding Remarks

This review aimed at discussing the data available that support the notion that presynaptic nerve endings can learn and memorize *acute* chemical stimulations. 

The functional and structural adaptations at the basis of this form of “chemically-induced presynaptic memory” develop either “in vitro”, after the *acute* exposure of synaptosomes to the triggering chemical, or “in vivo”, by *acutely* administering animals with the drug. 

In both cases, adaptations were retained beyond the removal of the triggering agents and emerged subsequently as changes in release efficiency and/or receptor-mediated events in “in vitro” experiments. 

The discovery of this chemically induced long-lasting memory expands our knowledge of the functional role of the presynaptic component of chemical synapses, also paving the road to new investigations concerning the impact of *acute* drug administration on neurotransmission. The finding that nerve endings can memorize chemical stimuli and that this presynaptic adaptation can emerge in “in vitro” studies open the possibility to evaluate the impact of the drugs in either physiological or pathological conditions, allowing also to identify drug-induced effects that do not necessarily imply a direct control of transmitter exocytosis. As to this regard, it is worth reminding that in recent years, attention was given to the mechanisms of “metamodulation”. The term of metamodulation refers to the possibility that different neurotransmitters can cooperate to modulate transmitter exocytosis in the CNS. Actually, although chemicals are often analyzed individually for their effects, the possibility that they can prime nerve endings and indirectly modify the release efficiency at nerve terminals as well as the expression and functions of the presynaptic receptors in plasmamembranes increases hugely the complexity of the scenario and must be taken into account when considering the impact of the occasional co-administration of therapeutics. Similarly, as environmental stimulation can control central transmission, rearing animals in enriched conditions as well as their stressful handling could cause acute central adaptation that might account for experimental variability.

Based on the data described in this review, the “chemically-induced presynaptic memory” elicited by *acute* exposure to drugs offers, on one hand, the possibility to investigate the cascades of events tuning synaptic transmission, but, on the other hand, it also unveils unexpected events leading to side effects due to anomalous adaptation of synaptic terminals and/or presynaptic release-regulating receptors that must be taken into consideration when evaluating the pharmacological profile of the receptors targeted at chemical synapsis.

## Figures and Tables

**Figure 1 ijms-20-03641-f001:**
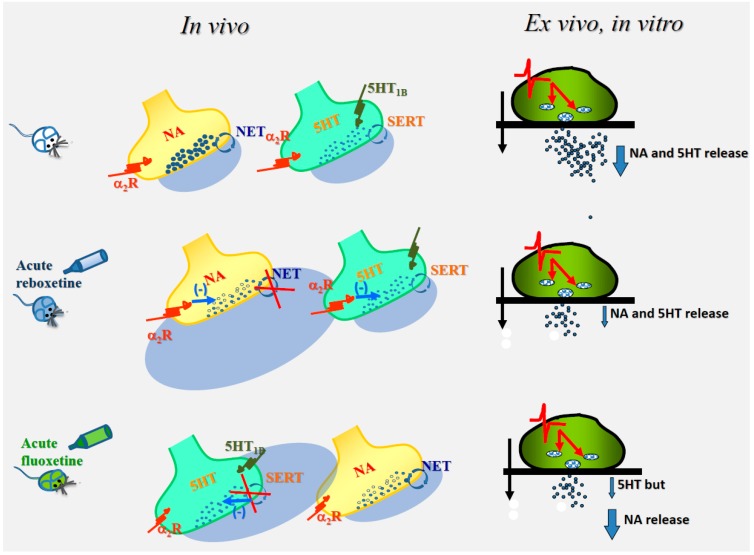
The cartoon describes the mechanisms proposed to trigger the “ex-vivo, in vitro” modifications of noradrenaline and serotonin exocytosis (blue arrow) in hippocampal synaptosomes isolated from animals *acutely* treated with reboxetine (middle) or fluoxetine (bottom) when compared to control condition (untreated animals, top). Noradrenergic terminals possess α_2_ autoreceptors, while serotonergic receptors possess both 5HT_1B_ autoreceptors and α_2_ heteroreceptors. It is proposed that the “in vivo” activation of both the α_2_ autoreceptors and the α_2_ heteroreceptors that would follow the *acute* administration of reboxetine might account for the reduced exocytosis of both noradrenaline and serotonin, while, based on notion that noradrenergic terminals do not possess serotonergic release-regulating heteroreceptors (left side of the Figure), serotonin-induced activation of serotonergic receptors in fluoxetine-treated animals only can occur at serotonergic terminals. The functional adaptations elicited by the “in vivo” activation of the presynaptic release-regulating receptors are retained by isolated synaptosomes and emerge subsequently in “in vitro” studies as changes in release efficiency (right side of the panel). Reboxetine and fluoxetine block the NA transporter (NET) and the 5HT transporter (SERT, circular arrow), increasing NA and 5HT bioavailability in the synaptic cleft. Synaptosomes were labelled with radioactive tracers and superfused with superfusion medium (black arrow). Synaptosomes were transiently (90 s) exposed to a depolarizing stimulus (the red line indicating the change in membrane potential) that causes vesicular exocytosis (red arrow) of transmitters (black dots).

**Table 1 ijms-20-03641-t001:** In vitro” persistent “presynaptic adaptation” in synaptosomes.

CNS Region	Transmitter	Drug Pre-Treatment	Depolarizing Stimulus Applied	Pretreatment Output	Reference
Rat hippocampus	Endogenous glutamate	PDE type V inhibitor PKA inhibitor	25 mM KCl for 30 s	⇓	[11]
Rat Nucleus Accumbens	[^3^H]dopamine	Nicotine (30 µM)	100 µM NMDA	⇓	[32,55]
Rat Nucleus Accumbens	[^3^H]dopamine	Nicotine (30 µM)	100 µM AMPA	⇓	[32,54]
Rat hippocampus	[^3^H] aspartate	CXCL12 (3 nM)	30 µM NMDA/3 nM CXCL12	⇓	[60,61]
Rat hippocampus	[^3^H] noradrenaline	CXCL12 (3 nM)	100 µM NMDA/3 nM CXCL12	⇓	[60,61]

The table summarizes the results from “in vitro” experiments, showing a persistent adaptation of synaptosomes elicited by exposing them transiently to enzyme modulators as well as to agonists of presynaptic receptors. Synaptosomes were superfused with a Kreb’s ringer containing the drug indicated in the third column and then exposed to a mild depolarizing stimulus (see the fourth column) to monitor the efficiency of glutamate exocytosis. The results clearly unveiled a significant reduction of transmitter exocytosis in those synaptosomes that were pre-exposed to the drugs when compared to control. ⇓= reduced.

**Table 2 ijms-20-03641-t002:** “ex vivo, in vitro” persistent “presynaptic adaptation” in synaptosomes.

In Vivo Treatment		“Ex Vivo, In Vitro” Synaptosomes Adaptations				
*Drug*	*Treatment*	Synaptosomal Preparation	Transmitter	Stimulus Applied	Outcome	Ref.
reboxetine	10 mg/Kg, os	rat hippocampus	[^3^H]noradrenaline	12 mM KCl	⇓	[8]
reboxetine	10 mg/Kg, os	rat hippocampus	[^3^H]serotonin	12 mM KCl	⇓	[8]
desipramine	10 mg/Kg, i.p.	mice cortex	[^3^H]noradrenaline	12 mM KCl	⇓	[71]
Desipramine-yohimbine	10 mg/Kg i.p. 0.5 mg/Kg i.p.	mice cortex	[^3^H]noradrenaline	12 mM KCl	No effect	[71]
fluoxetine	10 mg/Kg, i.p.	Rat hippocampus	[^3^H]serotonin	12 mM KCl	⇓	[8]
fluoxetine	10 mg/Kg, i.p.	Rat hippocampus	[^3^H]noradrenaline	12 mM KCl	No effect	[8]
LY379268	1 mg/Kg, i.p.	Mouse spinal cord	[^3^H]D-aspartate	15 mM KCl	⇓	[76]

The table summarizes the results from “ex vivo, in vitro” experiments showing a persistent adaptation in synaptosomes elicited by “in vivo” *acute* administration of drugs that modulate the functions of presynaptic release-regulating receptors. It is proposed that the *acute* administration of receptor ligands or carrier modulators elicit functional remodeling of the synaptic proteasome. The persistent “in vivo” adaptive changes in synaptic boutons are retained by nerve endings, after isolation from the hippocampus and washing with drug-free solutions, and emerge in “ex vivo, in vitro” study as reduced exocytosis efficiency. ⇓= reduced.

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
