# Peer review of "Acute Functional Adaptations in Isolated Presynaptic Terminals Unveil Synaptosomal Learning and Memory"

_ijms, 2019, doi:10.3390/ijms20153641_

Reviewer 1 Report

Review of " Functional adaptation in isolated presynaptic  terminals: do synaptosomes learn and memorize?"

In the introduction or the discussion, it might be good to comment on that maybe synaptosomes are not pure each time they are isolated. Thus, the variation which is observed among isolation even if from the same region of brain tissue is different.

Major points:

Introduction:

1. " In a whole, these findings allow the conclusion that synaptosomes can “learn and memorize” acute stimulations "

I see how the author would like to state “learn and memorize” but personally I think these terms also refer to generally a process involved with gene regulation and/or protein synthesis. However, if AMPA receptors can be inserted and removed from the membrane quickly to correspond to LTP/LTD then maybe such phenomenon can be implied to occur in synaptosomes and just state the synaptosomes maintain the cellular process, but not gene or protein synthesis. As Bailey et al., (2003) reported years ago.    If the intact neurons show LTP, LTD, and STF would one not expect some of the changes to also occur in the isolated tissues? If electrical activity can internalize or put proteins into the cell membrane, I would expect this could happen in isolated terminals.  However, we might expect many inconsistencies in synaptosomes as compared to intact circuits.

 I could see the Introduction being worded as these might be expected findings and that the points needed to be documented. But one should also comment on the properties are not fully able to be replicated when it comes to gene regulation and/or protein synthesis as for intact slices or NMJs. This is generally mentioned in the last sections of the report but disadvantages of using synaptosomes are not highlighted specifically in a section.

The last paragraph is a nice wrap up of the major points of the paper in that when using synaptosomes one does have to be cautious of the treatment effects and maybe even how the animals handled prior to sacrifice.

 Minor:

1. Abstract

" In particular, in 2003, Bailey and coll. Demonstrated"…..is it supposed to be Coll ?

Also not listed in reference list as far as I can tell.

 2. Abstract

 Not sure why capitalized here

 "..chemical Long Term Depression"

3.page 2 line 85 " tiddue preparation" I think this was meant to be  "tissue preparation"

 4. Line 179 ….paralleled by “ex-vivo, in vitro” analysis carried out whit synaptosomes isolated from the CNS ….

Typo

 5. Line 183 " and structural adaptations at the presynaptic component of the synapsis. The {THIS} results suggested …"

 6. Table 2

" acute administration administration of receptor ligands"

Seems odd to have administration listed twice. Also, spelling is off on 2nd time.

 7. In the schematic graphs showing the differences in release of transmitter with drug treatments. The pharmacological treatments might not be specific as assumed as even fluoxetine can likely block electrically evoked release in intact neurons. (Comparative Biochemistry and Physiology- Part C 176–177:52–61)

 8. There are various grammatical typos and some words merged together in the PDF but maybe that was a conversion issue to a PDF.

Author Response

Answers to REVIEWER 1’ s comments

 Review of " Functional adaptation in isolated presynaptic  terminals: do synaptosomes learn and memorize?"

 In the introduction or the discussion, it might be good to comment on that maybe synaptosomes are not pure each time they are isolated. Thus, the variation which is observed among isolation even if from the same region of brain tissue is different.

We thank the referee for the suggestion., Sentences have been introduced in the paragraph 3, concerning the variability of the preparation due to the isolation process (lines 117-120) as well as to the presence of  glial particles (lines 121-126).

Major points:

Introduction:

1. " In a whole, these findings allow the conclusion that synaptosomes can “learn and memorize” acute stimulations "

I see how the author would like to state “learn and memorize” but personally I think these terms also refer to generally a process involved with gene regulation and/or protein synthesis.

I agree with the referee that the term “learn and memorize” refers to events that originate from several pathways including the genetic ones. The review however discusses results obtained with synaptosomes that were acutely exposed to chemical stimuli. As far as the synaptosomes are concerned, the possibility that these particles possess the machinery required for the genetic expression of proteins and /or its regulation through epigenetic processes is still matter of discussion. However, to the best of my knowledge, direct evidence of the genetic synthesis of new proteins and its epigenetic regulation in synaptosomes are lacking (as already stated on line 274-276). Furthermore, also assuming that synaptosomes contain the genetic machinery to control proteins synthesis,  the timing of the synaptosomal treatment in “in vitro” experiments (20 minutes of exposure to the triggering chemical, the washout to re-equilibarte the system and then the exposure to a depolarizing agents to release transmitters) seems not sufficient to cause genetic / epigenetic modifications that might account for the synaptosomal adaptations. Finally, as to the acute “in vivo” treatment, the possibility that drug administration could have caused epigenetic and / or genetic changes in receptor expression deserves consideration. To the best of my knowledge, however, this possibility was not so far investigated.

Based on the referee’s comment, the term epigenetic / genetic was used only when discussing the impact of the “in vivo” (i.e. both chronic and acute ) drug administration (lines 46-51, lines 408-411)

However, if AMPA receptors can be inserted and removed from the membrane quickly to correspond to LTP/LTD then maybe such phenomenon can be implied to occur in synaptosomes and just state the synaptosomes maintain the cellular process, but not gene or protein synthesis.

Data exists in the literature showing that AMPA receptors as well as NMDA receptors traffic in-out synaptosomal plasma membranes in a constitutive manner (i.e. an in-out movements that develops in few minutes). The  possibility that synaptosomal adaptations could rely on changes in the in-out movements of these receptors deserves therefore attention and this possibility has been largely analyzed in the paper Grilli et al., 2012 (numbered 54 in the review) and it is now discussed on lines 303-312.

As Bailey et al., (2003) reported years ago.    If the intact neurons show LTP, LTD, and STF would one not expect some of the changes to also occur in the isolated tissues? If electrical activity can internalize or put proteins into the cell membrane, I would expect this could happen in isolated terminals.  However, we might expect many inconsistencies in synaptosomes as compared to intact circuits.

The referee is correct. The changes that underlie LTP and LTD would also occur in isolated tissues, as indeed demonstrated by Bayley and colleagues. Similarly, it is conceivable to hypothesize that electric activity would modify the trafficking of receptors in synaptosomes, although this hypothesis cannot be easily verified because of experimental limitations (the exposure of synaptosomes to an electric potential would kill them). As far as the inconsistencies observed in synaptosomes when compared to more intact preparation, these could depend on the presence of the biophase and of bystander cells in intact tissue when compared to synaptosomes in superfusion. This aspect has been already discussed in reviews dedicated to synaptosomes (see references n°13 and 14) and will no discuss further in this work.

I could see the Introduction being worded as these might be expected findings and that the points needed to be documented. But one should also comment on the properties are not fully able to be replicated when it comes to gene regulation and/or protein synthesis as for intact slices or NMJs. This is generally mentioned in the last sections of the report but disadvantages of using synaptosomes are not highlighted specifically in a section.

The referee is correct. The aim of this review was to document the observations that lead to the proposal of the hypothesis that synaptosomes can “learn and memorize”. I have been working with synaptosomes for more or less 40 years, but despite the experience coming from this training, I was impressed by the results that in the last decade unveiled the form of the synaptosomal plasticity that is matter of discussion of this review. The manuscript was not dedicated to comment the limitation of the use of synaptosomes when compared to other more intact preparations (i.e. slices and /or NMJs) or in general to describe their disadvantage with respect to other preparations. These aspects have been already discussed in previous reviews (see references n°13 and 14) and do not represent a topic of this review.

The last paragraph is a nice wrap up of the major points of the paper in that when using synaptosomes one does have to be cautious of the treatment effects and maybe even how the animals handled prior to sacrifice.

A sentence has been introduced on lines 494-496 to stress the impact of environmental stimulation on synaptosomal plasticity

 Minor:

1. Abstract

" In particular, in 2003, Bailey and coll. Demonstrated"…..is it supposed to be Coll ?

Also not listed in reference list as far as I can tell.

I apologize for the lack of accuracy. The abstract has been largely re-written and the reference omitted

 2. Abstract

 Not sure why capitalized here

 "..chemical Long Term Depression"

Again I apologize for the lack of accuracy

3.page 2 line 85 " tiddue preparation" I think this was meant to be  "tissue preparation"

Again I apologize for the lack of accuracy, the statement has been corrected

4. Line 179 ….paralleled by “ex-vivo, in vitro” analysis carried out whit synaptosomes isolated from the CNS ….

Typo

 I apologize for the lack of accuracy

5. Line 183 " and structural adaptations at the presynaptic component of the synapsis. The {THIS} results suggested …"

 Again I apologize for the lack of accuracy

6. Table 2

acute administration administration of receptor ligands"

Seems odd to have administration listed twice. Also, spelling is off on 2nd time.

I apologize for the lack of accuracy

7. In the schematic graphs showing the differences in release of transmitter with drug treatments. The pharmacological treatments might not be specific as assumed as even fluoxetine can likely block electrically evoked release in intact neurons. (Comparative Biochemistry and Physiology- Part C 176–177:52–61)

A note of caution has been introduced in the text on lines 356-357 to support the notion that mechanisms other that the presynaptic mechanism of regulation of serotonin release could be implicated in the effect observed.

8. There are various grammatical typos and some words merged together in the PDF but maybe that was a conversion issue to a PDF.

I apologize for the lack of accuracy, the manuscript has been largely revised

Reviewer 2 Report

This review by Pittaluga describes synaptosomes and presents studies to support the notion that synaptosomes change in response to chronic and acute drug exposure.

Major concerns.

The scope of the review is very biased, only presenting work in support of the hypothesis that isolated synaptosomes undergo plasticity. Further, it is not helpful that at least 21 of the 77 papers cited are also by the Pittaluga group. It is unclear whether this review adequately represents the consensus of the field at large. In addition, the title of the review poses the question "do synaptosomes learn and memorize". The readers would benefit from presenting studies that did not show similar results and discussion as help understand conflict.

 In several instances sweeping statements are made that are not well-explained or justified (e.g. lines 80-87). Dissecting pre- and post-synaptic components of the synapse has been performed since the 1950s at the neuromuscular junction. It is unclear, aside from just stating it, why the synaptosome is "one of the most suitable models". If the review makes the case that the synaptosome is a suitable model to study the presynaptic component, than the readers will benefit from discussion as to whether findings in synaptosomes have translated to more intact preparations.

It is unclear what is the most exciting, and recent work using synaptosomes. The reference that is highlighted (Bailey et al., 2003) is more than 15 years old. The review read more historical than reviewing recent studies.

Minor

The section on pathology is out of place. It is not described in the abstract or introduction.

There are many spelling errors. The manuscript needs proofreading.

Author Response

Answers to REVIEWER 2’s comments

 Comments and Suggestions for Authors

This review by Pittaluga describes synaptosomes and presents studies to support the notion that synaptosomes change in response to chronic and acute drug exposure.

Major concerns.

The scope of the review is very biased, only presenting work in support of the hypothesis that isolated synaptosomes undergo plasticity. Further, it is not helpful that at least 21 of the 77 papers cited are also by the Pittaluga group. It is unclear whether this review adequately represents the consensus of the field at large.

The referee’s feeling is correct, the review is dedicated to support the hypothesis that synaptosomes can “learn and memorize” acute stimuli. I started to propose this hypothesis in 2007 and during the last decade I dedicated part of my research to verify it. For this reason, the data supporting this hypothesis mainly originates from my laboratory. On the other hand it is worth stressing that only few laboratory works on superfused synaptosomes to study transmitter release and its control mediated by presynaptic receptors. Honestly speaking I carefully checked out  the literature concerning drug treatment/ synaptosomes / presynaptic receptor / transmitter release when I started to work to the review and I did it again right now, before answering your comments, but I was not able to find works from other groups concerning the topic of this manuscript , i.e. the impact of acute “in vitro” and / or “in vivo”  treatments on synaptosomes functions. I am perfectly aware that this represents an important limitation of the review, but I cannot tackle the problem differently. I am also aware of the fact that an important percentage of references are mine papers, but they have been cited just to support the “working hypothesis” developed during time.

Finally, the hypothesis proposed is in a sense new and it is difficult that it already gained consensus. Rather I thought it was fundamental to propose it for the first time, just to open the discussion on this aspect.

 In addition, the title of the review poses the question "do synaptosomes learn and memorize". The readers would benefit from presenting studies that did not show similar results and discussion as help understand conflict.

As stated above, I checked again the literature in order to find studies showing results that would contrast the proposed hypothesis but I could not find data that contrast the hypothesis and that could help discussion 

 In several instances sweeping statements are made that are not well-explained or justified (e.g. lines 80-87). Dissecting pre- and post-synaptic components of the synapse has been performed since the 1950s at the neuromuscular junction. It is unclear, aside from just stating it, why the synaptosome is "one of the most suitable models". If the review makes the case that the synaptosome is a suitable model to study the presynaptic component, than the readers will benefit from discussion as to whether findings in synaptosomes have translated to more intact preparations.

The manuscript has been largely revised to avoid sweeping statements.

It is unclear what is the most exciting, and recent work using synaptosomes. The reference that is highlighted (Bailey et al., 2003) is more than 15 years old. The review read more historical than reviewing recent studies.

As already stated the manuscript deals with results starting from the first work of Bailey et al., 2003 till the more recent from my lab (Di Prisco et al., 2013 and 2016) and therefore could be also considered an historical review.

 Minor

The section on pathology is out of place. It is not described in the abstract or introduction.

The section concerning the pathology has been now described in the abstract. The section has been modified and it is still reported in the text since I believe that provide informations that could be of interest to the comprehension of the proposed hypothesis.

There are many spelling errors. The manuscript needs proofreading.

I apologize for the lack of accuracy. The manuscript has been deeply revised for spelling and typos.

 Reviewer 3 Report

Professor Pittaluga belongs to the research group that originally developed the synaptosomal preparation maintained in vitro and described most of the physiologic and pharmacological characteristics of presynaptic receptor/carrier under normal circumstances. The proposed reappraisal of the literature concerned with the use of this nerve ending preparation in vitro demonstrates its important role in any multidisciplinary approach to study synaptic transmission under normal and pathological conditions. The studies carried out in experimental models of nervous diseases and after in vivo acute and chronic pharmacological treatment together with the dissection of the plastic changes maintained in vitro underscore the translational importance of the preparation that will provide us with unexpected and fundamental information in the years to come.

Typos to be corrected:

page 2, line 47: “…that strictly dependent..” should read “.. strictly dependent..”

page 2, line 85: “...a tiddue…” probably should be “...a tissue..”

page 7, line 300: “…events that underlying….” should read “… events underlying..”

Author Response

Answers to REVIEWER 3’s comments

Comments and Suggestions for Authors

Professor Pittaluga belongs to the research group that originally developed the synaptosomal preparation maintained in vitro and described most of the physiologic and pharmacological characteristics of presynaptic receptor/carrier under normal circumstances. The proposed reappraisal of the literature concerned with the use of this nerve ending preparation in vitro demonstrates its important role in any multidisciplinary approach to study synaptic transmission under normal and pathological conditions. The studies carried out in experimental models of nervous diseases and after in vivo acute and chronic pharmacological treatment together with the dissection of the plastic changes maintained in vitro underscore the translational importance of the preparation that will provide us with unexpected and fundamental information in the years to come.

 Typos to be corrected:

 page 2, line 47: “…that strictly dependent..” should read “.. strictly dependent..”

 page 2, line 85: “...a tiddue…” probably should be “...a tissue..”

 page 7, line 300: “…events that underlying….” should read “… events underlying..”

 I thank the referee for the positive comments and for the consideration of the work of my laboratory. The paper has been largely revised to improve the quality and to correct spelling and typos.

Round  2

Reviewer 2 Report

The author has addressed my concerns. However, the title posed as a questions still suggests that both sides of the argument will be presented and should be changed.

Author Response

Answer to the second report of the Reviewer #2

The title has been changed and in the proposed new form it does not pose question, but it proposes a conclusion that it is proved by the results discussed in the review. Hoping that in the new version the title is acceptable for publication, best regards

Anna Pittaluga